# Survival after Pure (Acute) Erythroid Leukemia in the United States: A SEER-Based Study

**DOI:** 10.3390/cancers15153941

**Published:** 2023-08-03

**Authors:** Kriti Gera, Daniela Martir, Wei Xue, John R. Wingard

**Affiliations:** 1Department of Medicine, University of Florida College of Medicine, Gainesville, FL 32610, USA; kriti.gera@ufl.edu (K.G.); daniela.martirvargas@medicine.ufl.edu (D.M.); 2Department of Biostatistics, University of Florida College of Medicine, Gainesville, FL 32610, USA; wxue@cog.ufl.edu; 3Division of Hematology and Oncology, Department of Medicine, University of Florida College of Medicine, Gainesville, FL 32610, USA

**Keywords:** pure erythroid leukemia, acute myeloid leukemia, overall survival, chemotherapy, hypomethylating agents

## Abstract

**Simple Summary:**

Pure erythroid leukemia (PEL), a rare subtype of acute myeloid leukemia (AML), is characterized by the proliferation of malignant erythroid precursors. To understand the status of PEL at the population level, a retrospective analysis was conducted using the Surveillance Epidemiology and End Results (SEER) database from 2000 to 2019. The study included 968 patients with a confirmed diagnosis of PEL, with a median age of 68 years and a higher proportion of males (62%). Among the patients, 62.5% received chemotherapy. The overall survival (OS) rates varied significantly based on age, with better outcomes seen in younger age groups. Chemotherapy was associated with improved OS both in adults and children. However, there were no significant differences in OS based on sex, race, ethnicity, or median household income. Over the two decades of the study, there was no notable improvement in OS, indicating poor outcomes with current chemotherapeutic agents. The study underscores the urgent need for investigational agents that are capable of inducing remission.

**Abstract:**

Background: Acute erythroid leukemia (AEL), also known as pure erythroid leukemia, is a rare subtype of acute myeloid leukemia (AML) characterized by the proliferation of malignant erythroid precursors. Outcome data at the population level are scarce. Methods: We performed a retrospective analysis of the Surveillance Epidemiology and End Results (SEER) database. All cases with a histologically confirmed diagnosis of acute (pure) erythroid leukemia during the period of 2000–2019 were included in the study. The Kaplan–Meier method was used to perform survival analysis. The significance of differences between overall survival (OS) was analyzed using the log-rank test. Results: In total, 968 patients were included in the study. The median age was 68 years (range 0–95), 62% of patients were males, and 62.5% (*n* = 605) were treated with chemotherapy. The median OS for <18, 18–49, 50–64, 65–79 and 80+ age groups was 69, 18, 8, 3 and 1 month, respectively (*p* < 0.0001). Patients who received chemotherapy had significantly improved OS compared to patients who did not, among both adults (*p* < 0.0001) and children (*p* = 0.004). There were no significant differences in OS based on sex, race, ethnicity and median household income. Median OS for adults diagnosed in 2000–2004, 2005–2009, 2010–2014, 2015–2019 was 4, 6, 6 and 3 months, respectively, with no significant differences in OS between these groups. Conclusion: AEL occurs in all age groups but is most common in the elderly. Outcomes are poor with current chemotherapeutic agents, with no improvement in the last two decades. This study stresses the urgent need for investigational agents.

## 1. Introduction

Pure erythroid leukemia (PEL), also known as acute erythroid leukemia (AEL), is categorized as a distinct subtype of acute myeloid leukemia (AML) under the 2016 and 2022 World Health Organization (WHO) classification of myeloid neoplasms and acute leukemias [1]. It accounts for approximately 1% of all cases of AML [2]. The current WHO diagnostic criteria for PEL require more than 80% of bone marrow cells to be of erythroid lineage with at least 30% proerythroblasts [1]. Furthermore, nearly all cases of PEL exhibit TP53 immunohistochemical positivity, which serves as the basis for its designation within the broad category of AML with mutated TP53 by the International Consensus Classification. PEL can occur at any age but is typically seen in adults.

Patients with PEL typically present with pancytopenia [3], with the red cell and platelet cell lines being the most commonly affected. The peripheral blood smear may reveal various nonspecific morphologies of cells, including schistocytes, immature erythrocytes and pseudo-Pegler–Huet neutrophils. The diagnosis is established through a bone marrow biopsy. The bone marrow core biopsy usually shows increased cellularity with abundant sheets of leukemic PEL cells. These PEL cells are typically large with centrally located round nuclei, one or more nucleoli, and basophilic cytoplasm [4]. Multilineage dysplasia is often observed. There is no specific immunophenotypic marker for PEL, although erythroid-specific markers such as glycophorin A, hemoglobin and spectrin may be weakly present or absent [5,6]. These factors, coupled with the complex karyotype, make diagnosis challenging, even for the most experienced pathologists [7]. Moreover, it is imperative to exclude other similar morphological presentations, such as other subtypes of AML, pernicious anemia and hemolytic anemias, as these are potentially treatable conditions with favorable prognoses.

PEL can develop de novo but is more frequently associated with prior therapy and other hematological neoplasms [8,9,10]. Erythroblastic sarcoma is an extremely rare extramedullary presentation of PEL, with only a few cases reported in the literature [11,12]. PEL is universally characterized by complex cytogenetics, TP53 mutations [13], and poor response to standard AML/MDS therapies [14]. These features are present irrespective of whether PEL develops de novo or is secondary in nature [8]. The most commonly observed chromosomal abnormalities are deletions of chromosomes 5/5q and 7/7q [15]. Like other subtypes of AML, pediatric PEL is considered as a separate entity to adult PEL, with distinct molecular features and pathogenesis [16,17]. However, due to the infrequency of this disease in the pediatric population, genetic and molecular characteristics are not well characterized. In a study of pediatric AEL cytogenetic characteristics, four out of five pediatric PEL patients exhibited a complex karyotype. Three patients had NUP98 fusions, while two had a KDM5A fusion [16]. Additionally, five case reports of pediatric PEL have documented NFIA with core binding factor gene rearrangements, suggesting a potentially distinct pathological subgroup of pediatric PEL [18,19,20,21,22]. Mutations commonly found in other subtypes of AML, such as FLT3, CEBPA and NPM1 are usually not observed in either adult or pediatric PEL, signifying a distinct origin.

PEL portends an aggressive course and likely dismal outcome. Due to a low prevalence and its aggressive course, with poor overall prognosis, there is a paucity of data on the management of PEL. Current treatment strategies include the use of intensive chemotherapy, hypomethylating agents (HMA), such as azacitidine and decitabine, and allogeneic bone marrow transplants (BMTs). Over the past decade, several studies have found hypomethylating agent regimens to be equivalent or superior with regards to patients’ overall survival (OS) [23,24,25]. A recent study at the Mayo Clinic using the current classification schemas showed no benefit of one treatment approach over another [13]. BMT is currently only recommended once complete remission is achieved. However, a significant number of patients unfortunately succumb to the disease before attaining remission through upfront treatments.

The diagnostic criteria for AEL have undergone changes over the last decade. In the 2008 WHO classification, AEL was subdivided into two subtypes: M6a (bilineage AEL), which included both erythroid precursors and myeloid blasts, and was later classified as MDS (myelodysplastic syndrome) or AML with myelodysplasia-related changes (AML-MRC) in the 2016 WHO classification; and M6b, also known as Pure erythroid leukemia, which is characterized by presence of blasts of erythroid lineage and remained classified as subtype of AML in the 2016 classification. As a result, both PEL and erythroleukemia were classified as two subtypes of the broader AEL designation, essentially leading to the inclusion of what is now viewed as a type of myelodysplastic syndrome (MDS) in the analysis of cases of PEL [26]. Consequently, population level data on PEL are scarce [24]. This retrospective study was conducted to understand the current status of PEL in the United States, according to the contemporary classification, with a focus on demographic characteristics and survival outcomes over the past two decades.

## 2. Material and Methods

### 2.1. Data Source and Study Population

The 17 Population-Based Registries Research Plus Data in the National Cancer Institute Surveillance Epidemiology and End Results (SEER) database from 2000–2019 (November 2021 Submission) were accessed for this study (SEER 2021) [27]. The SEER 17 covers approximately 34.6% of the population in the USA and consists of data available from 2000–2019 from 17 Registries (San Francisco-Oakland SMSA, Connecticut, Hawaii, Iowa, New Mexico, Seattle (Puget Sound), Utah, Atlanta (Metropolitan), San Jose-Monterey, Los Angeles, Alaska Natives, Rural Georgia, California excluding SF/SJM/LA, Kentucky, Louisiana, New Jersey, and Greater Georgia). The data were collected through the SEER*Stat v8.4.0.1 software package. Patients were identified using the International Classification of Disease for Oncology 3rd Edition [ICD-O-3] code for PEL: 9840. All patients (pediatric and adult) diagnosed between 2000 and 2019 with histologically confirmed PEL were included in this study.

### 2.2. Study Variables

The demographic characteristics of PEL patients extracted from the SEER database included age at diagnosis, sex, race, ethnicity, median household income, survival status, and survival time. All variables were present for all patients included in the analysis. The race variable included White race, Black race and other race subgroups. The ethnicity variable included Hispanics and Non-Hispanics. Patients were further subdivided into two groups based on age (age < 18 years and age ≥ 18 years). Patients were divided into four groups based on their period of diagnosis (2000–2004, 2005–2009, 2010–2014 and 2015–2019) to reflect the effects of different treatment modalities. All the patients reported between 2000 and 2019 in the SEER database were included in this study, and the target event was death by any cause. The survival time was the time interval since the date of diagnosis to either death date or last follow-up date for those still alive in 2019. Subjects who were still alive by the end of follow-up in 2019 were considered censored for the event. Censoring is a statistical procedure that appropriately handles individuals who remain alive at the end of the study period, ensuring their inclusion and accounting for their ongoing status in the analysis [28].

### 2.3. Statistical Analysis

Statistical analyses were performed using SAS software version 9.4. Descriptive statistics were used to summarize baseline demographic characteristics. Survival data were calculated as the duration between the date of diagnosis and the last follow-up or death from any cause. OS was estimated using the Kaplan–Meier method. The log-rank test was used to compare survival amongst groups. The Shapiro–Wilk test was used to test for normality of the sample data. Hartigan’s dip statistics were used to test for multimodality. All statistical tests were two-sided and *p*-values < 0.05 were considered statistically significant.

## 3. Results

### 3.1. Patient Characteristics

A total of 968 patients satisfied the inclusion criteria and were included in the study. The median age of the study population was 68 years (range 0–95, mean 64 years). The age distribution of the study cohort, dichotomized based on gender, is shown in Figure 1. The majority of the patients were >60 years (70.87%) with male (61.98%) and non-Hispanic white (72%) predominance. The distribution of age ranges deviated significantly from normality (W = 0.8779, *p* < 0.0001). The test for multimodality using Hartigan’s dip statistics rejected the assumption of unimodality with a statistically significant *p*-value of 0.012, indicating the presence of multiple modes in the distribution. Baseline characteristics of the cohort are outlined in Table 1.

Among the children and adolescent group, 49 out of 50 patients were included for survival analysis as one patient did not have survival information. Of these patients, 44% were non-Hispanic whites and 55% were males. Among adults, 905 out of 918 patients were included for analysis as 13 patients did not have survival information. Of these, 73.7% were non-Hispanic whites, 62.8% were males and 61.7% (*n* = 559) were treated with chemotherapy.

### 3.2. Survival

Survival was significantly worse with advancing age (*p* < 0.0001) (Figure 2). The median OS for <18, 18–49, 50–64, 65–79 and 80+ years age groups was 69, 18, 8, 3 and 1 month, respectively. The 5-year survival rates for <18, 18–49, 50–64, 65–79 and 80+ years age groups were 55.0%, 32.9%, 13.0%, 4.2% and 2.2%, respectively. For the purposes of this study, patients were divided into two age groups: age < 18 years (children and adolescent group) and age ≥ 18years (adult group). Median OS and 5-year survival for different demographic groups has been presented in Table 2 and Table 3. Of children alive at 1 year (*n* = 37), 62% (*n* = 23) were alive at 5 years, suggesting that there is a plateau for many patients of this age group. In contrast, of adults alive at 1 year (*n* = 285), only 24.9% (*n* = 71) were alive at 5 years, suggesting few are cured by current therapies.

#### 3.2.1. Children and Adolescents

Among the children and adolescents, there was no significant difference in survival between groups based on race, ethnicity, median household income and year of diagnosis (Appendix A). Median OS for females and males was 152 and 18 months, respectively (*p* = 0.1111) (Appendix A). There was significantly improved survival in patients who received chemotherapy vs. those who did not. (Median OS of 152 months vs. 2 months, *p* = 0.0004) (Appendix A).

#### 3.2.2. Adults

Among the adult groups (age ≥ 18), there were no significant differences in survival between groups based on race, ethnicity and median household income (Appendix A). The median OS for female and males was 4 and 5 months, respectively (*p* = 0.5064) (Appendix A). The median OS for groups based on the year of diagnosis, i.e., 2000–2004, 2005–2009, 2010–2014 and 2015–2019, was 4, 6, 6 and 3 months, respectively (*p* value 0.0824) (Appendix A). There was significantly improved survival in patients who were treated with chemotherapy compared to those who were not (median OS of 8 months vs. 1 month, *p* < 0.0001) (Figure 3), although OS was poor even with chemotherapy, suggesting that those who did not receive chemotherapy were diagnosed at an advanced stage or were suffering from serious comorbidities.

## 4. Discussion

PEL is a rare and aggressive hematological neoplasm characterized by uncontrolled expansion of erythroid precursors, primarily proerythroblasts. Proerythroblasts play a crucial role in resistance to treatment, and increased frequency of proerythroblasts is associated with a poorer prognosis [29]. This study represents the largest population analysis in the United States over the past two decades, providing insights into the demographic characteristics and outcomes of Acute (Pure) Erythroid Leukemia. SEER (Surveillance, Epidemiology, and End Results) provides extensive coverage of approximately 48% of the U.S. population, including population-based cancer registries in 22 geographic areas. As a result, the data are highly representative of the entire U.S. population demographics, enabling accurate insights into cancer incidence and survival rates for diverse populations across the country [30]. Additionally, each SEER registry is reviewed by a facilitator, with a cutoff for agreement on diagnostic information prior to inclusion in the database. This careful evaluation process helps to maintain the accuracy and reliability of the data within the SEER database.

PEL can manifest at any age. The distribution of age often appears to be bimodal, with a smaller peak for individuals under 20 years old and a more prominent and broader peak for people in the seventh decade of life [31]. In our study, only a small percentage (5%) of cases were observed in patients under 18 years old. The incidence increased after the age of 40 and peaked in the 70–80 years age range. However, it was observed that the distribution did not exhibit a unimodal pattern (Figure 1). Although studies have found clear male predominance, our study showed a male to female ratio of 55:4 [32]. Like other studies, we also found a higher proportion of Hispanic whites.

PEL is associated with poor prognosis, with median survival typically being less than 6 months. Previous studies have reported dismal outcomes. For instance, a study at the Mayo Clinic, with inclusion of 29 PEL cases (mean age 66 years, range 27–86), reported a median overall survival of 1.8 months and 1-year survival of 0% [13]. Similarly, a study of 15 cases (median age 68 years, range 47–87) of de novo PEL reported a median OS of 1.4 months [33]. A study of 22 cases (median age 69, range 37–81) of PEL reported a median OS of 2.8 months [8]. All these studies included patients over 25 years of age, which may have contributed to the poorer OS than our study. In contrast, a large multinational study of 217 patients reported median OS of 11 months with 1-year survival of 49%. This study also included patients with MDS and AML-MRC, based on the WHO 2008 classification. Previous smaller studies to compare outcomes between M6a (erythroleukemia) and M6b (PEL) subtypes have reported strikingly lower survival rates in PEL [34]. Similarly, our study also reported a median OS of 5 months and 1-year survival of 26.8% (adult PEL) which is much worse than the previously reported survival rates of AEL with inclusion of both M6a and M6b subtypes. Moreover, our study highlights that, despite advancements in transplant procedures, blood banking, and the approval of novel therapies, the survival rates for PEL have not shown improvement over the past two decades. Additionally, even though there was no significant difference observed among different age groups, the study revealed a worse median overall survival (OS) in later years. This decline in OS may be attributed to the continuous advancements in pathology and the evolving definition of PEL over time. The diagnostic criteria for PEL have become more stringent, leading to the exclusion of patients who do not strictly meet the newly defined criteria. In the WHO 2008 classification, PEL was characterized by the expansion of primitive erythroblasts (including proerythroblasts and immature cells in earlier stages), comprising at least 80% nucleated cells in the bone marrow. However, with the updated WHO 2016 classification, an additional requirement of at least 30% proerythroblasts was introduced, making the criteria for diagnosing PEL more restrictive [4]. Due to this evolution in the definition of PEL, some cases that would have been diagnosed as PEL under the previous criteria may no longer meet the updated standards. Consequently, the number of reported PEL cases is evidently less in 2015–2019 than in other groups. Moreover, the frequency of proerythroblasts has been identified as having prognostic value, which may explain the poorer observed outcomes in those latter years. Another possible explanation may be due to a greater proportion of cases of secondary AML and the emergence of chemotherapy-resistant clones over time [35]. Secondary AML arises as a result of prior exposure to cytotoxic therapies or as an evolution from a pre-existing hematological disorder, and it constitutes a significant proportion of PEL cases. According to SEER data, the incidence of therapy related AML (t-AML) has increased from 0.04/100,000 people in 2001–2007 to 0.2/100,000 people in 2008–2014 [36]. In a recent study involving 22 cases of PEL, it was found that only 23% (five cases) had a de novo origin, while 50% of the cases were therapy-related and 23% had a prior history of MDS [8]. The median OS for de novo PEL was 3.9 months compared to 2.3 months for those with therapy related PEL [8]. Secondary PEL is particularly resistant to standard induction treatments, highlighting the need for novel strategies to overcome this chemotherapy resistance and improve outcomes.

Pediatric PEL demonstrates distinctive genetic features and a presumedly different pathogenesis compared to that of adults. As evident in our study, it is a rare condition in children but is associated with favorable outcomes. However, when compared with the erythroid/myeloid subtype, it is important to note that PEL is associated with worse outcomes even in children. In a study involving 24 cases (median age 10.2, range, 0.5–21 years), of which five patients had PEL, the 5-year survival rate was observed to be 20% in the PEL group, while the erythroid/myeloid group had a significantly higher 5-year survival rate of 66% [16]. In the CCG-2891 pediatric MDS/AML trial, a 5-year survival rate of 31.6% was observed for AEL (M6) subtype [37]. However, this study included both erythroid/myeloid and pure erythroid subtypes within the group. Furthermore, the study revealed that pediatric MDS, M6, and M7 subtypes had poorer survival outcomes compared to other subtypes of AML. The major cause of that finding was induction failure. However, it was observed that when patients with these subtypes achieved remission, their survival rates were similar to those with other subtypes of AML. This is in line with our study finding that the survival curve is characterized by a plateau, likely representing stabilization of survival for many patients once they successfully achieved remission. The presence of this plateau is seen in all age groups, but is particularly prominent in children, possibly because of their inherent life expectancy.

There is no standard treatment for PEL. In our cohort, both pediatric and adult groups showed significantly improved survival with chemotherapy. However, it is worth noting that many patients with PEL expire before receiving treatment, which may have led to skewed findings. In the adult groups, while treatment with chemotherapy demonstrated significantly improved survival, both treatment groups overall displayed poor survival. This suggests that the patients who did not receive chemotherapy might have presented with an advanced stage of the disease or had significant comorbidities. Current consensus for treatment is intensive chemotherapy, hypomethylating agents and allogeneic BMT. Hypomethylating agents are generally used in patients who are unable to tolerate intensive chemotherapy or have been heavily pretreated with chemotherapeutic agents. In 2018, the combination of azacitidine/decitabine or low-dose cytarabine with venetoclax received approval for the treatment of newly diagnosed AML patients who are not suitable candidates for intensive chemotherapy [38]. In a multinational study involving 291 patients with TP53 mutated AML, the combination of azacitidine/decitabine or low-dose cytarabine with venetoclax demonstrated the highest success rate in achieving complete remission (CR) [39]. Furthermore, the study revealed that overall survival was dependent on the response to the initial induction therapy. A Mayo Clinic study of 41 PEL patients comparing different treatment strategies, viz., HMA alone, HMA plus venetoclax, intensive chemotherapy and best supportive care, no approach revealed better outcomes than the other. In our study, 26% (243) of adult patients survived at 1 year, out of which 29% (71) made it to the 5 year mark. From these studies, one could infer that while consolidative treatments such as allogeneic BMT may help in sustaining long term remissions, the need for novel agents and drug combinations capable of inducing remission is pressing and requires immediate attention. Understanding the pathogenesis of PEL is challenging due to its complex karyotype and heterogeneous mutational spectrum. Additionally, the rarity of the disease and diagnostic challenges further complicate the development of new therapies. However, these distinctive features offer potential opportunities for development of novel targeted therapies.

There are several limitations to consider in this study. The retrospective nature of the study and the unavailability of information on specific treatment modalities and etiologies (de novo vs. secondary PEL) restricts the ability to make conclusive statements about treatment efficacy. Although SEER is a reliable and representative data source, there may be a potential bias based on differential healthcare accessibility and regional disparities despite the large numbers and representativeness.

## 5. Conclusions

In conclusion, outcomes of PEL remain dismal with no significant improvement in survival over the last 20 years. This study underscores the need for new investigational agents that are capable of inducing remission. Further studies are warranted to understand the pathogenesis and molecular characteristics of PEL, which may aid in the development of therapeutic strategies.

## Figures and Tables

**Figure 1 cancers-15-03941-f001:**
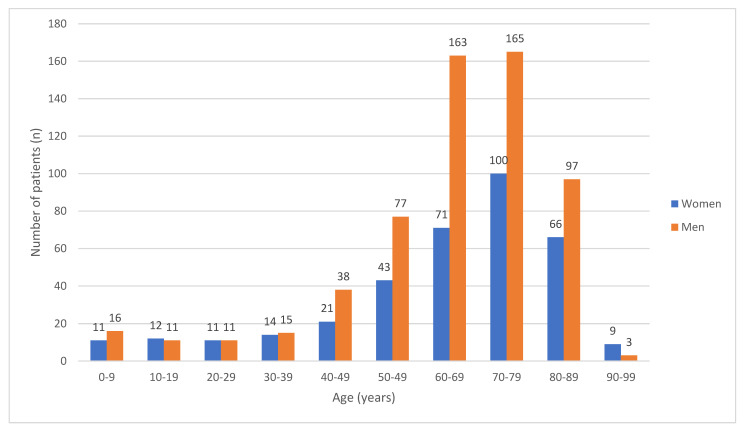
Age distribution dichotomized based on gender for patients (*n* = 968) who met inclusion criteria.

**Figure 2 cancers-15-03941-f002:**
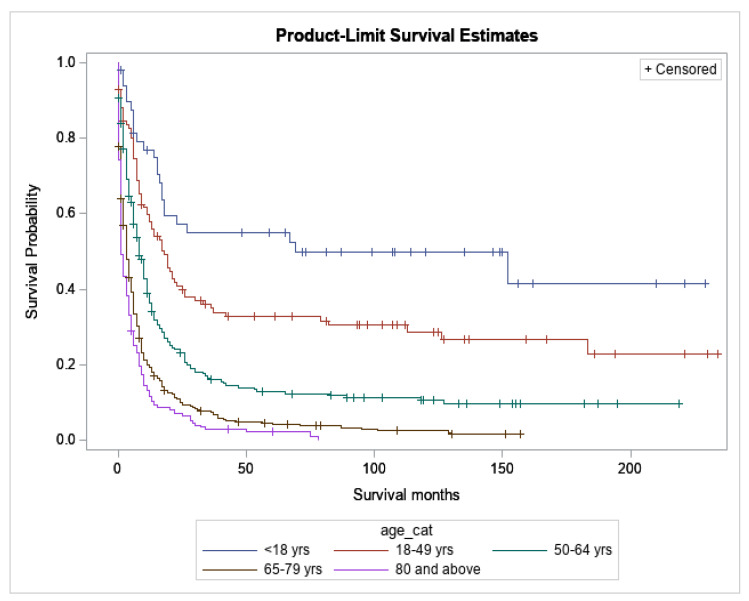
Overall Survival based on age (*p* < 0.0001).

**Figure 3 cancers-15-03941-f003:**
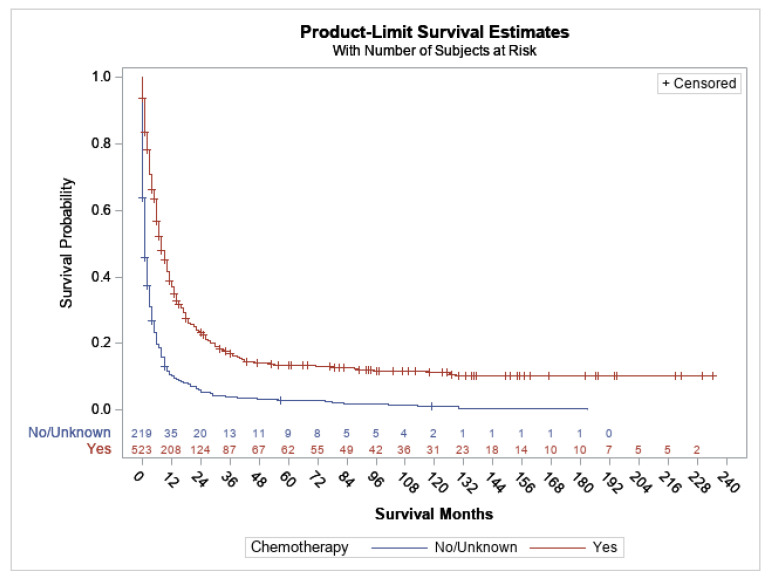
Overall survival based on treatment with chemotherapy for PEL (age ≥ 18 years), *p* < 0.0001.

**Table 1 cancers-15-03941-t001:** Baseline characteristics of patients with PEL included in the study (*n* = 968).

Baseline Characteristics	Number of Patients (Age < 18 Years)	Number of Patients (Age ≥ 18 Years)
Sex		
Female	23	573
Male	27	345
Race and Ethnicity		
Hispanic	17	91
Non-Hispanic Black	5	67
Non-Hispanic White	22	675
Non-Hispanic Asian or Pacific Islander	6	78
Non-Hispanic American Indian/Alaska Native	0	4
Non-Hispanic Unknown Race	0	3
Median household income		
<$75,000	42	582
≥$75,000	8	336
diagnosis year		
2000–2004	7	237
2005–2009	22	246
2010–2014	12	270
2015–2019	9	165
chemotherapy		
No/unknown	4	359
yes	46	559

**Table 2 cancers-15-03941-t002:** Median OS for different demographic groups with Pure Erythroid Leukemia.

	Median OS (in Months)	
	Adults (Age ≥ 18 Years)	Children and Adolescent Group (Age < 18 Years)
Overall	5	69
Non-Hispanic White	5	*
Non-Hispanic Unknown Race	*	*
Non-Hispanic Black	6	17
Non-Hispanic Asian or Pacific Islander	6	16
Non-Hispanic American Indian/Alaska Native	7	*
Hispanic (All Races)	4	*
Household Income = <$75,000	5	152
Household Income = ≥$75,000	4	16.5
Chemotherapy = No/Unknown	1	2
Chemotherapy = Yes	8	152
Female	4	152
Male	5	18
Diagnosis year 2000–2004	4	*
Diagnosis year 2005–2009	6	23
Diagnosis year 2010–2014	6	*
Diagnosis year 2015–2019	3	18

* Undefined; Median OS was not reached in this group at the last follow up date.

**Table 3 cancers-15-03941-t003:** 5-year survival for different demographic groups with Pure Erythroid Leukemia.

	5-Year Survival (%)	
	Adults (Age ≥ 18 Years)	Children and Adolescent Group (Age < 18 Years)
Overall	9.47	55.01
Non-Hispanic White	8.93	57.27
Non-Hispanic Black	6.20	40.00
Non-Hispanic Asian or Pacific Islander	15.13	33.33
Non-Hispanic American Indian/Alaska Native	0.00	*
Hispanic (All Races)	10.47	66.08
Household Income = <$75,000	9.64	58.71
Household Income = ≥$75,00	9.17	37.50
Chemotherapy = No/Unknown	2.74	0.00
Chemotherapy = Yes	13.62	58.74
Female	11.71	67.61
Male	8.10	44.32
Diagnosis year 2000–2004	9.09	85.71
Diagnosis year 2005–2009	10.86	43.32
Diagnosis year 2010–2014	9.05	66.67
Diagnosis year 2015–2019	*	*

* Undefined: no subjects to measure 5-year survival.

## Data Availability

The data that support the findings of this study are publicly available from https://seer.cancer.gov/.

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
