# Peer review of "Survival after Pure (Acute) Erythroid Leukemia in the United States: A SEER-Based Study"

_cancers, 2023, doi:10.3390/cancers15153941_

Round 1

Reviewer 1 Report

The research paper titled "Demographic Characteristics and Outcomes of Acute (Pure) Erythroid Leukemia: A Population Study" provides a comprehensive analysis of the clinical features, diagnosis, treatment, and outcomes of Pure Erythroid Leukemia (PEL). The study focuses on a large population of 968 patients with PEL and aims to shed light on this rare and aggressive hematological neoplasm.

The strengths of the study lie in its large sample size and detailed examination of various aspects of PEL. The authors conducted a thorough analysis of the demographic characteristics of the patient population, including age distribution and gender ratio. They observed a bimodal age distribution, with a smaller peak in patients under 20 years and a larger peak in the seventh decade. The study also revealed a higher male to female ratio and a higher proportion of Hispanic whites among the PEL patients.

The authors discuss the diagnostic challenges associated with PEL, highlighting the lack of specific immunophenotypic markers and the need to differentiate it from other similar morphological presentations. They emphasize the importance of accurate diagnosis through bone marrow biopsy and the exclusion of other potentially treatable conditions. Furthermore, the authors discuss the genetic and molecular features of PEL, such as complex cytogenetics and TP53 mutations, which contribute to poor response rates to standard AML/MDS therapies.

Regarding survival outcomes, the study reports a median overall survival of 5 months and a 1-year survival rate of 26.8% for adult PEL cases, indicating poor prognosis. However, pediatric PEL cases showed better outcomes compared to adults, suggesting potential differences in the disease characteristics and response to treatment between age groups. The authors acknowledge the rarity of PEL in the pediatric population and the need for further research to better understand its genetic and molecular characteristics in children.

Treatment approaches for PEL are discussed, including intensive chemotherapy, hypomethylating agents, and allogeneic hematopoietic stem cell transplant. While chemotherapy showed improved survival in both pediatric and adult cases, overall survival rates remain unsatisfactory. The authors stress the need for new investigational agents and targeted therapies to improve outcomes for PEL patients.

The study has a few limitations worth noting. The retrospective nature of the research and the lack of information on specific treatment modalities and etiologies (de novo vs. secondary PEL) limit the ability to draw definitive conclusions about treatment efficacy. Additionally, the study's focus on a population-based analysis in the United States may introduce biases related to healthcare access and regional variations.

In conclusion, the research paper provides valuable insights into the demographic characteristics, diagnostic challenges, treatment approaches, and survival outcomes of PEL. It highlights the need for further research to understand the genetic and molecular characteristics of PEL, as well as the development of new therapies. Despite its limitations, this study contributes to the existing knowledge on PEL and serves as a basis for future investigations in this field.

 The language used in the research paper is appropriate for the target audience of healthcare professionals, researchers, and scholars in the field of hematology and oncology. 

Author Response

The authors thank the reviewer for positive comments regarding the structure, results and analysis, and purpose of the manuscript. 

Reviewer 2 Report

Acute erythroid leukaemia, also known as pure erythroid leukaemia, is a very rare subtype of acute myeloid leukaemia (AML), approximately 1% of all AML.  What is more important, despite significant progress in treatment, especially in young AML patients, in the erythroid leukaemia the outcome is still poor.  

This is one of the biggest epidemiology data, including almost 1,000 people treated from 2000 to 2019.  

The work is very readable, however, in such a large analysis, one may feel a lack of information regarding, in particular, treatment, and in particular those patients who managed to achieve remission. 

In the Supplements Figure D, the author should at least try to discuss the differences in OS, especially that 2015-2019 have to worse outcome  

Figure 4 is redundant. As in any leukaemia, the lack of treatment results in the quick death of the patient. 

best regards

Author Response

The authors thank the reviewer for positive comments regarding the scope and utility of the paper as well as constructive feedback about providing more information and enhancing the results. A point-by-point response is listed below. 

Comment 1: The work is very readable, however, in such a large analysis, one may feel a lack of information regarding, in particular, treatment, and in particular those patients who managed to achieve remission. In the Supplements Figure D, the author should at least try to discuss the differences in OS, especially that 2015-2019 have to worse outcome.

Authors: Authors thank reviewer for their valuable suggestion to include possible explanations for worse survival in later years. The text has been amended as follows, “Additionally, even though there was no significant difference observed among different year groups, the study revealed a worse median overall survival (OS) in later years. This decline in OS may be attributed to the continuous advancements in pathology and the evolving definition of PEL over time. The diagnostic criteria for PEL have become more stringent, leading to the exclusion of patients who do not strictly meet the newly defined criteria. In the WHO 2008 classification, PEL was characterized by the expansion of primitive erythroblasts (including proerythroblasts and immature cells in earlier stages), comprising at least 80% of nucleated cells in the bone marrow. However, with the updated WHO 2016 classification, an additional requirement of at least 30% pro-erythroblasts was introduced, making the criteria for diagnosing PEL more restrictive4. Due to this evolution in the definition of PEL, some cases that would have been diagnosed as PEL under the previous criteria may no longer meet the updated standards. Consequently, the number of reported PEL cases are evidently lesser in 2015-19 than other groups . More-over, the frequency of proerythroblasts has been identified as having prognostic value, which may explain the observed worse outcomes in those latter years. Another possible explanation may be due to a greater proportion of cases of secondary AML and the emergence of chemotherapy-resistant clones over time35. Secondary AML arises as a result of prior exposure to cytotoxic therapies or as an evolution from a pre-existing hematological disorder, and it constitutes a significant proportion of PEL cases. According to SEER data the incidence of therapy related AML (t-AML) has increased from 0.04/100,000 people in 2001-2007 to 0.2/100,000 people in 2008-201436. In a recent study involving 22 cases of PEL, it was found that only 23% (5 cases) had a de novo origin, while 50% of the cases were associated with therapy-related and 23% had prior history of MDS8. The median OS for de novo PEL was 3.9 months compared to 2.3 months for those with therapy related PEL8. Secondary PEL is particularly resistant to standard induction treatments, highlighting the need for novel strategies to overcome this chemotherapy resistance and improve outcomes. (Page 9 paragraph 2 , lines 266-292).

Information regarding treatment and remission rates was not available in our dataset. The text has been amended to include cited literature of these aspects.

Comment 2: Figure 4 is redundant. As in any leukemia, the lack of treatment results in the quick death of the patient. 

Authors: Figure 4 has been removed.

Reviewer 3 Report

Estimated Authors,

I've read with great interest the present report on a relatively rare blood hematological neoplasm (Pure Erythroid Leukemia). In this retrospective, database-based, study, Gera et al. were able to report and discuss about a total of 968 patients. According to their analyses, Age ≥ 18 yrs, Male gender, no previous chemotherapy, were associated with a more dismal prognosis.

The study is quite interesting, but several conceptual and formal improvements are, unfortunately, required.

First of all, a methodological issue is represented by the characteristics of your study population. According to Figure 1, your study included a total of 52 (i.e. around 5%) of individuals aged < 18 yrs, 113 aged 20 - 50 yrs, and the large majority of participants was therefore aged > 50 yrs. Moreover, around 30% of participants was aged 80 yrs or more. As a consequence, your statistical analysis should take in account the heterogeneity of your sample and the potential differences in survival that should be acknowledged as disease-independent. More precisely, as 1/3 of your sample is (at recruitment) aged more than 80 years, their actual survival may be reduced not only because of the disease, but because their life expectancy according to parent population group was over. As a consequence: 1) please discuss how censoring may have influenced your data (https://www.ncbi.nlm.nih.gov/pmc/articles/PMC3932959/); 2) please provide analyses based on more age groups (e.g. < 18, 18 - 49, 50 - 64, 65 - 79, 80+) whose design should more accurately reflect your study population avoiding to collapse individuals aged less than 50 yrs with older ones; 3) provide a 5-year analysis (alive vs. death) by factors included in Table 1.

Second: discussion is not totally consistent with the data you provided and reported. Your data do not provide any information about topic such as markers, mutations, etc. Interesting as they clearly are, they exceed the scope of this study and should be both simplified and more properly focused on the main text.

Third: please discuss the reliability of the data you provided as they were derived from a quite reliable source (but their representativity should be discussed and accurately assessed).

Finally, please fix some issues in tables and figures:

Figure 1: age group cannot be 0-10, 10-20 and so on, but rather 0-9, 10-19, etc; Please dichotomize the reporting by gender of included cases.

Figure 2 and following. Please double check the number of decimal figures, particularly in p values: all reported decimal figures should be consistent across the main text (i.e. 3 or 4 figures, but consistently).

Author Response

The authors thank the reviewer for critical, actional feedback regarding the quality, organization, results, and analyses presented within the paper which certainly will enhance the overall purpose and clarity of the manuscript. Point-by-point responses are listed below.

Comment 1. First of all, a methodological issue is represented by the characteristics of your study population. According to Figure 1, your study included a total of 52 (i.e. around 5%) of individuals aged < 18 yrs, 113 aged 20 - 50 yrs, and the large majority of participants was therefore aged > 50 yrs. Moreover, around 30% of participants was aged 80 yrs or more. As a consequence, your statistical analysis should take in account the heterogeneity of your sample and the potential differences in survival that should be acknowledged as disease independent. More precisely, as 1/3 of your sample is (at recruitment) aged more than 80 years, their actual survival may be reduced not only because of the disease, but because their life expectancy according to parent population group was over. As a consequence: 1) please discuss how censoring may have influenced your data (https://www.ncbi.nlm.nih.gov/pmc/articles/PMC3932959/); 2) please provide analyses based on more age groups (e.g. < 18, 18 - 49, 50 - 64, 65 - 79, 80+) whose design should more accurately reflect your study population avoiding to collapse individuals aged less than 50 yrs with older ones; 3) provide a 5-year analysis (alive vs. death) by factors included in Table 1.

Authors: The authors thank the reviewer for valuable suggestions and the opportunity to describe the study methodology. 1) The text has been amended as follows, “Censoring is a statistical procedure that appropriately handles individuals who remain alive at the end of the study period, ensuring their inclusion and accounting for their ongoing status in the analysis.” (lines 129-132), 2) The requested analysis is provided in lines 170-173 “The median OS for <18, 18-49, 50-64, 65-79 and 80+ years age groups was 69, 18, 8, 3 and 1 month, respectively.5-year survival rates for <18, 18-49, 50-64, 65-79 and 80+ years age groups was 55.0%, 32.9%, 13.0%, 4.2% and 2.2% respectively.” and new Figure 2. Original Figure 2 has been removed. 3) Table 3 has been added to provide a 5-year analysis for the factors included in Table 1.

Comment 2. Second, Discussion is not totally consistent with the data you provided and reported. Your data do not provide any information about topic such as markers, mutations, etc. Interesting as they clearly are, they exceed the scope of this study and should be both simplified and more properly focused on the main text.

Authors: Authors thank the reviewer for valuable feedback.  The content originally present in lines 299-325, which included information about markers, mutations, and pathogenesis, has been relocated from the discussion section and condensed within the Introduction section of the manuscript as suggested by reviewer (highlighted in green).

Comment 3. Third: please discuss the reliability of the data you provided as they were derived from a quite reliable source (but their representativity should be discussed and accurately assessed).

Authors: The authors thank Reviewer 3 for their comments regarding inclusion of the reliability of the SEER database. The text has been amended to include information about the data retrieval process employed by SEER registries as follow: “SEER (Surveillance, Epidemiology, and End Results) provides extensive coverage of approximately 48% of U.S. population including population-based cancer registries in 22 geographic areas. As a result, the data are highly representative of the entire U.S. demographics, enabling accurate insights into cancer incidence and survival rates for diverse populations across the country. Additionally, each SEER registry is reviewed by a facilitator at each SEER registry with a cutoff of agreement on diagnostic information prior to inclusion in the database. This careful evaluation process helps maintain the accuracy and reliability of the data within the SEER database.” (Lines 230-238).

Comment 4. Figure 1: age group cannot be 0-10, 10-20 and so on, but rather 0-9, 10-19, etc; Please dichotomize the reporting by gender of included cases.

Authors: Figure 1 has been amended to classify age and dichotomized for gender as suggested by the reviewer.

Comment 5. Figure 2 and following. Please double check the number of decimal figures, particularly in p values: all reported decimal figures should be consistent across the main text (i.e. 3 or 4 figures, but consistently).

Authors: Text has been amended for consistency of 4 decimal figures.

Reviewer 4 Report

In my opinion the level of novelty of this study is too low to allow the publication in prestigious Cancers. What the authors did was a simple collection of the publicly available data and simple analysis. The discussion is very limited and mostly descriptive-but no other discussion can be done when the amount of new results is so low. From the technical point of view the study is acceptable, but the authors should choose another journal to present such a routine work. My detailed comments can be found below.

In the title, the Authors suggest that this is a “review of literature”. This manuscript is for sure not a review, first because it is too short, the number of references is limited, it doesn’t follow the PRISME indications for review article, etc. This is misleading.

Lines 36-37, „can occur” is repeated

Line 46, precisely what hypomethylating agents are being used? Azacitidine? Decitabine? Other?

Line 56, this “previous classification” should be described

Line 73, at some places the Authors write “Pure Erythroid Leukemia” while in the rest “Pure erythroid leukemia” – this must be unified

Lines 86-87, how many subjects were still alive?

Line 99, were all the necessary data available for all of the 968 subjects?

Line 93, why nonparametric test and not more statistically powerful (parametric) test has been used? I guess it was because of the violation of some (what?) assumptions.

Figure 1, is this data distributed normally? Have you used Shapiro-Wilk test to check it?

Table 1, was Median household income included as a variable in this study? It is not listed in the Methods section 2.3. If not, why is it presented in this table?

Line 117, it should be Figure 2, not Figure 1

Line 120, what kind of plateau do you have in mind?

Table 2, what was the Median overall survival (in months) for Children and adolescent group with Diagnosis year 2010-2014 ?

The quality (resolution) of Figure 2 must be improved

Lines 178-179, is this in accordance with the results of this study?

Lines 164-176, this is not a discussion but an introduction

Lines 181-182, actually there are statistical tests that can be used to prove if the distribution is bimodal. The Authors, however, haven’t done this kind of analysis.

Lines 190-224, again, this so-called discussion has nothing to do with the present analysis and is not a discussion but an introduction

Finally, what I really miss is some kind of the meta-analysis.

Author Response

The authors thank the reviewer for their comments regarding the rigor and novelty of the manuscript, which have been addressed in revisions to enhance the quality of the work. Point-by-point responses are provided below. 

Comment 1. In the title, the Authors suggest that this is a “review of literature”. This manuscript is for sure not a review, first because it is too short, the number of references is limited, it doesn’t follow the PRISME indications for review article, etc. This is misleading.

Authors: The title has been amended such that “review of literature” was removed.

Comment 2. Lines 36-37, „can occur” is repeated.

Authors: Repeated phrase has been removed.

Comment 3. Line 46, precisely what hypomethylating agents are being used? Azacitidine? Decitabine? Other?

Authors: The text has been amended as follows: “Current treatment strategies include the use of intensive chemotherapy, hypomethylating agents (HMA) such as azacitidine and decitabine, and allogeneic stem cell transplant.” (Line 80).

Comment 4: Line 56, this “previous classification” should be described (described in lines 65-72)

Authors: The text has been amended as follows: “The diagnostic criteria for AEL have undergone changes over the last decade. In the 2008 WHO classification, AEL was subdivided into two subtypes; M6a (bilineage AEL), which included both erythroid precursors and myeloid blasts, and was later classified as MDS (myelodysplastic syndrome) or AML with myelodysplasia-related changes (AML-MRC) in the 2016 WHO classification. M6b, also known as Pure erythroid leukemia, is characterized by presence of blasts of erythroid lineage and remained classified as subtype of AML in the 2016 classification.  As a result, both PEL and erythroleukemia were classified as two subtypes of the broader AEL designation, essentially leading to the inclusion of what is now viewed as a type of myelodysplastic syndrome (MDS) in the analysis of cases of PEL.” (Lines 89-98).

Comment 5. Line 73, at some places the Authors write “Pure Erythroid Leukemia” while in the rest “Pure erythroid leukemia” – this must be unified.

Authors: Text amended for consistency in this phrase where appropriate.

Comment 6.

Lines 86-87, how many subjects were still alive?

Authors: 115 patients were alive at the end of follow up period.

Comment 7. Line 99, were all the necessary data available for all of the 968 subjects?

Authors: All variables were present for all patients included in the analysis, text amended to clarify this point (Line 120).

Comment 8. Line 93, why nonparametric test and not more statistically powerful (parametric) test has been used? I guess it was because of the violation of some (what?) assumptions?

Authors: We chose logrank test to compare survival distribution amongst groups because logrank test is the most commonly used statistical test for comparing the survival distributions of two or more groups. The log-rank test is most powerful under proportional hazards (PH). Under non-proportional hazards (non-PH), the log-rank test loses power, but it is still statistically valid under non-PH. In this study there are  large number of adult patients which mitigate the power loss. The PH assumption is not rejected in younger aged patients thus log-rank test does not lose substantial power under this case.

Comment 9. Figure 1, is this data distributed normally? Have you used Shapiro-Wilk test to check it?

Authors: Shapiro-Wilk test was performed. The distribution of age ranges deviated significantly from normality (W=0.8779, p <0.0001). The text has been amended to add this information (Line 150-151).

Comment 10. Table 1, was Median household income included as a variable in this study? It is not listed in the Methods section 2.3. If not, why is it presented in this table?

Authors: Household income was a variable in the study, and text has been amended to clarify this omission as follows: “The demographic characteristics of PEL patients extracted from the SEER database included age at diagnosis, sex, race, ethnicity, median household income, survival status, and survival time.” (Line 119).

Comment 11. Line 117, it should be Figure 2, not Figure 1.

 Authors: Figure label has been amended as suggested.

Comment 12. Line 120, what kind of plateau do you have in mind?

Authors: A plateau in survival curves refers to a period in which the survival rates of a particular group of individuals remain relatively stable or unchanged over a certain period of time. In this case, we are referring to the stabilization of survival after 1-year in children and adolescent which signifies favorable response to treatment in achieving long term survival. The text was amended to discuss this point in the Discussion section as follows, “This is in line with our study findings as the survival curve is characterized by a plateau, likely representing stabilization of survival for many patients, once they successfully achieved remission. The presence of this plateau is seen in all age groups, but is particularly prominent in children, possibly because of their inherent age expectancy. (Lines 308-312) 

Comment 13. Table 2, what was the Median overall survival (in months) for Children and adolescent group with Diagnosis year 2010-2014?

Authors: It was undefined as median OS was not reached at the last follow-up date for these groups. Text amended to clarify this information.

Comment 14. The quality (resolution) of Figure 2 must be improved

Authors: Original Figure 2 has been removed. A new figure 2 was added to include suggested methodological changes by another reviewer. 

Comment 15. Lines 178-179, is this in accordance with the results of this study?

Authors: No, this is based on literature review, reference has been added. Text was amended to clarify study findings.

Comment 16. Lines 164-176, this is not a discussion but an introduction

Authors: Text was removed from discussion and condensed in the introduction as suggested by reviewer (highlighted in green in Introduction).

Comment 17. Lines 181-182, actually there are statistical tests that can be used to prove if the distribution is bimodal. The Authors, however, haven’t done this kind of analysis.

Authors: Test for multimodality was performed. The text has been amended as follows: “The test for multimodality using Hartigans’ dip statistics rejected the assumption of unimodality with a statistically significant p-value of 0.012, indicating the presence of multiple modes in the distribution”(Lines 150-153).

Comment 18. Lines 190-224, again, this so-called discussion has nothing to do with the present analysis and is not a discussion but an introduction.

Authors: Text was removed from discussion and condensed in the introduction as suggested by reviewer (highlighted in green in introduction) .

 Comment 19. Finally, what I really miss is some kind of the meta-analysis

Authors: The Title was revised to underline that this is not a literature review. The discussion has been amended to include a more comprehensive review of literature relevant to the study.

Round 2

Reviewer 3 Report

The paper has been accurately amended, therefore I'm happy to endorse its eventual acceptance.

Reviewer 4 Report

The Authors have included my comments in the revised version. Despite the low level of scientific novelty, I find this work acceptable for Cancers.